# Salivary Cortisol Reaction Norms in Zoo-Housed Great Apes: Diurnal Slopes and Intercepts as Indicators of Stress Response Quality

**DOI:** 10.3390/ani12040522

**Published:** 2022-02-21

**Authors:** Verena Behringer, Jeroen M. G. Stevens, Ruth Sonnweber

**Affiliations:** 1Endocrinology Laboratory, German Primate Center, Leibniz Institute for Primate Research, Kellnerweg 4, 37077 Göttingen, Germany; 2Behavioral Ecology and Ecophysiology, Department of Biology, University of Antwerp, Universiteitsplein 1, 2610 Antwerp, Belgium; jeroen.stevens@uantwerpen.be; 3SALTO, Odisee University of Applied Sciences, Hospitaalstraat 26, 9100 Sint-Niklaas, Belgium; 4Department of Behavioral and Cognitive Biology, University of Vienna, Djerassiplatz 1, 1030 Vienna, Austria; ruth-sophie.sonnweber@univie.ac.at

**Keywords:** animal welfare, bonobo, orangutan, gorilla, enrichment, non-invasive monitoring, glucocorticoid

## Abstract

**Simple Summary:**

Changes in cortisol fluctuations are used for stress monitoring. Methodologically, this is straight forward, because sample collection is simple and analytical methods advanced, but since cortisol is primarily a hormone that facilitates energy allocation, the interpretation of these measures is often complex. Acute elevations in cortisol levels are not per se bad, but may constitute an adaptive coping mechanism. Likewise, low cortisol levels do not always indicate the absence of a stressor. To distinguish between stress response qualities, a more fine-grained analyses of cortisol fluctuations is warranted. Cortisol excretion follows a diurnal pattern with high levels in the morning, decreasing throughout the day. Two regression coefficients describe this curve: the intercept (the constant with which levels change throughout the day) and the slope (curve steepness and direction). We assessed salivary cortisol intercepts and slopes in zoo-housed apes on routine days, enrichment days, and in the new ape house. While cortisol excretion increased on enrichment days, the daily decline of cortisol levels was not affected. The move to the new house seemingly was a major stressor as cortisol levels increased slightly and the circadian cortisol decrease was impaired. The combination of intercept and slope measures can differentiate between stress responses, thereby constituting a useful tool for stress monitoring.

**Abstract:**

Monitoring changes in cortisol levels is a widespread tool for measuring individuals’ stress responses. However, an acute increase in cortisol levels does not necessarily denote an individual in distress, as increases in cortisol can be elicited by all factors that signal the need to mobilize energy. Nor are low levels of cortisol indicative for a relaxed, healthy individual. Therefore, a more fine-grained description of cortisol patterns is warranted in order to distinguish between cortisol fluctuations associated with different stress response qualities. In most species, cortisol shows a distinct diurnal pattern. Using a reaction norm approach, cortisol levels across the day can be described by the two regression coefficients: the intercept and the slope of the curve. We measured immunoreactive salivary cortisol in three zoo-housed ape species under three conditions (routine days, enrichment days, and after the move to a new house). We examined salivary cortisol intercepts (SCI) and salivary cortisol slopes (SCS) of the diurnal curves. SCI and SCS were independent from each other. SCI was highest on enrichment days and lowest on routine days. SCS was steep on routine days and blunted after the move. Only SCI was species-specific. Our study provides evidence that combining SCI and SCS measures allows us to differentiate between types of stress responses, thereby constituting a useful tool for welfare assessment.

## 1. Introduction

For decades, the steroid hormone cortisol is used as a physiological marker to measure and monitor individuals’ responses to stressors [1,2]. This methodology is widely considered as the “gold standard” to study stress responses and animal welfare [3,4]. From a practical point of view, cortisol can be measured relatively easily in its native or metabolized form in various sample types, such as saliva, urine, feces, hair, or feathers. One big advantage of these kinds of samples is that they can be collected non-invasively, also in wild living animals. Depending on the substrate, instantaneous cortisol levels (e.g., in saliva) or cumulative levels of cortisol (and its metabolites) excretion (e.g., in feces) can be obtained [4,5], making cortisol measurement an attractive tool for stress monitoring. However, the interpretation of changes in cortisol levels is not quite that simple [6,7,8]. When a stressor is stimulating the hypothalamic-pituitary-adrenal (HPA) axis by threatening homeostasis, cortisol is secreted to restore homeostasis. Thus, an increase in cortisol levels reflects a threat to homeostasis, and, at the same time, an organism’s attempt to restore homeostasis [6,8]. This adaptive process of restoring homeostasis, resulting in changes in cortisol levels, can be elicited by all factors that signal the need to mobilize energy. These range from fluctuations in ambient temperature, food intake, activity patterns (e.g., increased locomotion, but also mating and reproduction), and external stressors, such as noise exposure and social conflicts [4,9,10]. Additionally, life-history (reproductive periods and events) and environmental factors (e.g., season) have an impact on cortisol levels [11,12,13]. Cortisol levels also increase as a response to pleasurable experiences, termed positive valence arousal [14].

In addition to these factors related to variation in cortisol levels, blood cortisol levels exhibit typical episodic fluctuations over a 24 h cycle, referred to as the diurnal rhythm of cortisol. In diurnal species, cortisol values are highest in the morning shortly after awakening, declining rapidly in the following hours. Blood cortisol levels reach a nadir in the first few hours of sleep and remain low during the night [15,16,17,18]. This diurnal pattern is widespread across different diurnal species, e.g., domestic animals (horses [19]), cetaceans (*Tursiops aduncus, Orcinus orca* [20]), elephants ((*Loxodonta africana* and *Elephas maximus* [21]), (*Elephas maximus* [22])), and apes (*Pan paniscus*) [23], and can be detected not only in blood, but also in other body fluids, such as saliva [19,22,23] and urine [21]. This rhythm has been linked to the function of diverse processes, such as sleep–wake-patterns e.g., [24], metabolism, learning, and memory, e.g., [25], or immune functions [26]. A disrupted cortisol pattern during the day can increase the likelihood of various diseases (reviewed in [27]). Therefore, the assessment of the diurnal cortisol pattern and whether it is disrupted is of utmost importance when assessing an animal’s physical health state and wellbeing.

Methodologically, there are different approaches to assess diurnal cortisol pattern. Most commonly, either the surge in cortisol concentrations around awakening (the cortisol awakening response), total cortisol output across the day, or properties of the diurnal cortisol curve across the day are evaluated [17]. The diurnal slope is defined “*as the line resulting from regression of cortisol values collected across the day, excluding the morning awakening response*” [17]. The diurnal cortisol decrease is linear over time. Therefore, it can be described by two regression parameters: the intercept and the slope of the cortisol level curve across the time of day [28]. In reaction norm approaches, individuals’ average responses to a mean centered environment (reaction norm intercept) and its level of plasticity (reaction norm slope) in response to that environmental gradient are examined [29,30,31]. Reaction norm repeatability then indicates how stable these responses are across environmental change over time. Although there is between individual variation in diurnal cortisol slopes [11], within healthy individuals the pattern itself seems very repeatable [32,33,34]. The interruption of this typically stable diurnal slope can be used as an indicator of a persistent stressor [17]. In humans, a negative diurnal slope is indicative of healthy HPA axis functioning, while positive or flattened diurnal slopes can be suggestive of a potential HPA axis dysfunction. Thus, diurnal cortisol slopes can be studied as mechanistic indicators linking stress levels with health—so far done mainly in humans [17]. In humans, cortisol slopes are commonly classified as inconsistent, typical, or flat, based on the steepness of cortisol decrease throughout the day [10,35]. For example, flatter cortisol slopes were found in humans with posttraumatic stress disorder [36], depression [37], fatigue [38], and a history of maltreatment [39,40]. Also in non-human animals, a blunted diurnal slope may be indicative of a long-term stressor: for example, the diurnal slope was blunted in barren housed pigs compared to enriched housed pigs [41]. Therefore, the diurnal cortisol pattern in combination with absolute levels of cortisol may allow us to distinguish between long-term/persistent (potential negative/detrimental) and short-term (positive/neutral) stress responses. Short-term stressors will result in a surge in cortisol excretion, but the decline of cortisol levels thereafter will not be impaired. A negative or a toxic stressor is a situation or an experience the individual is not able to cope with, with a great duration and magnitude, which may then result in a cortisol dysregulation or an allostatic overload [42,43].

Cortisol level changes are used to assess the impact of husbandry on stress states in zoo-housed great apes. Usually, cortisol levels are measured in hair, urine, saliva, feces, or blood. Salivary cortisol has been measured in all great ape species [44], and has been used in the context of intragroup competition [45,46,47], parturition [48], response to management events such as veterinary visits [49], or positive reinforcement training [50]. However, assessment of diurnal cortisol patterns, using reaction norm methodologies has not yet been undertaken in zoo-housed apes so far. We argue that applying a reaction norm approach to diurnal fluctuations in salivary cortisol can provide qualitative information, useful for better understanding and interpreting individual stress responses—as is already done in humans. By making use of our knowledge about the pattern of circadian salivary cortisol excretion, we can monitor deviations from this pattern (across the whole day or in specific time windows of the day) in relation to different conditions.

The aim of this study is to measure reaction norm salivary cortisol intercepts (SCI) and reaction norm salivary cortisol slopes (SCS) in three zoo-housed ape species: bonobos (*Pan paniscus*), western lowland gorillas (*Gorilla g. gorilla*), and Sumatran orangutans (*Pongo abelii*). We investigated these two parameters (SCI and SCS) in three different conditions: (i) on normal everyday life routine days (the old ape house), (ii) on enrichment days (enrichment devices were provided in the old ape house), and (iii) after the move to a new ape house with a completely new structure of environment and daytime routine (new ape house). In the first condition (old house), we expected to find the lowest SCI, and a typical non-disrupted decrease of the SCS in the old house, where the apes were habituated to their environment and daily routines, and therefore, experienced most predictability and maybe perceived control in this condition [51]. Thus, the old house is considered to be the “baseline” diurnal cortisol pattern. For the second condition (enrichment days), we expected SCI to be higher in comparison to routine days, because previous studies have shown that environmental enrichment can be associated with elevated cortisol levels in great ape species [52,53]. We also expected to find a typical SCS, similar to the old house pattern, as cortisol levels should be downregulated after the excitement of the enrichment exposure. In the third condition (new house), the apes were confronted with a new environment, different daily routines, and therefore, with a generally less predictable surrounding, which can also be seen as an enrichment but at the same time as a long-lasting stressor [54]. We predicted higher SCI and a disrupted, flattened SCS in the new house as compared to the old house.

## 2. Materials and Methods

### 2.1. Subjects

From November 2006 to December 2008, we studied three ape species at Zoo Frankfurt, Germany: (i) a group of twelve bonobos (one adult male, seven adult females, one immature female, and three immature males), (ii) a group of nine gorillas (one silverback, five adult females, one immature female, and two immature males), and (iii) a group of seven orangutans (one adult male, three adult females, two immature males, and one immature female). For most of the time of the study, the apes had access to their indoor enclosures (see the method section regarding the new house) at all times except during cleaning. Apes were provisioned with vegetables and fruits several times per day and had *ad libitum* access to fresh water.

### 2.2. Study Sections

The study was divided into three sections: (i) routine days in the “old ape house”, (ii) enrichment days (in the old ape house), and (iii) days in the “new ape house”.

(i)Old ape house: At the beginning of the study, all apes were housed in the original ape house (old ape house hereafter) that was constructed in 1933 and since then had been enlarged and modernized several times. Apes were used to management routines at the old ape house and familiar with the structure of their environment. In this building, the apes had access to indoor and outdoor areas, as well as separation boxes where they could retreat from the visitors view. Indoor and outdoor enclosures contained a combination of natural substrates like biofloor (gorilla indoor and one bonobo indoor), some concrete covered with epoxy coating, and metal climbing structures, as well as ropes and branches. All inside enclosures were separated from the visitor area with glass; in the outside enclosure this was either mesh (bonobos and orangutans) or glass (gorillas). The bonobo indoor enclosure consisted of three rooms, which could be separated from each other, (total surface 69 m^2^, height 3.5 to 7 m). The outside enclosure consisted of three parts (total surface of 85 m^2^). The gorilla inside enclosure was one room (total surface 93 m^2^, height 5 to 7 m). The gorilla outside enclosure measured (total surface 470 m^2^). The orangutan group was kept in three inside enclosures (total surface 42 m^2^, 3.7 m high) and three outside enclosures (total surface 86 m^2^). The enclosures contained a combination of natural substrates, some concrete covered with epoxy coating, and metal climbing structures.(ii)Enrichment days: In the old ape house, enrichment items were presented to the individuals of all three ape species. Boxes, hoses, or tennis balls were baited with treats and offered to the apes. All ape species were familiar with these forms of environmental enrichment, as it was provided regularly to the apes prior to this study. During our study, each of these familiar enrichment items was presented to each group on four consecutive days. Only one enrichment type (either box, or hoses, or tennis balls) was presented on a given day. In bonobos and orangutans, enrichment was provided from 1 p.m. to 4 p.m. and in gorillas, due to management reasons, from 11 a.m. to 4 p.m.(iii)New ape house: Between 13 May 2008 and 15 May 2008, all apes were moved to a new ape house (new ape house hereafter). This was an entirely novel environment to all individuals, and involved new management routines, such as food was prepared and coming from an unknown direction, some construction was ongoing, and keepers also had to adapt to the new situation. Additionally, during the first month after the move, the building was closed to visitors. Only indoor areas were accessible to the individuals during our data collection period. The indoor enclosure for the bonobos has two parts, (total surface 147 and 59 m^2^; 7–8 m high). The gorillas’ inside enclosure consists of two parts (total surface 389 m^2^). The orangutans have access to two inside enclosures (102 and 152 m^2^—height 10–12 m). All ape enclosures have bio-floor, artificial stone walls, and climbing structures are dead branches and tree trunks as well as ropes.

### 2.3. Saliva Sampling Protocol

For saliva sample collection, positive reinforcement techniques were used [50,55]. The apes were trained to chew on cotton rolls (Salivette R, Sarstedt, Nümbrecht, Germany) and hand them back to the experimenter. Individuals received food rewards for participating in this procedure, also during the study period. No samples were collected after a reward was given to the study subject.

Samples were collected between November 2006 and December 2008. In the old ape house, samples were collected between November 2006 and May 2008. Within this period, enrichment was provided and sampling of saliva on enrichment days took place. The apes were moved in May 2008 to their new ape houses. Saliva was then collected in the new ape house from May 2008 until the end of December 2008. The number of saliva samples per species and per condition is shown in the Table 1. For orangutans and bonobos, saliva sample collection took place twice per day, at 1 p.m. and at 4 p.m., both in the old and the new ape house. On enrichment days, four saliva samples were collected: one at 1 p.m. at the timepoint when enrichment was provided, a second and a third after 10 and 20 min after the enrichment device was presented, and a fourth one at 4 p.m. Due to management reasons, saliva sample collection from gorillas was undertaken at 11 a.m. and at 4 p.m. in all three conditions. One person took around seven minutes to collect saliva samples of all orangutans; in the other two species this took marginally longer.

Collected samples were stored at −20 °C until analysis at the University of Veterinary Medicine, Department of Biomedical Sciences, Vienna, Austria. After arrival in the lab, samples were thawed to room temperature, centrifuged (1500× *g*, 10 min), and analyzed for immunoreactive cortisol levels.

### 2.4. Sample Preparation and Analytical Methods

For immunoreactive cortisol measurement, a cortisol enzyme immunoassay (EIA) was used. The EIA and the cross-reactivities have previously been described by Palme and Möstl [56]. Samples were first diluted 1:10 with assay buffer. The assay had previously been validated for the measurement of salivary cortisol in apes [57]. The EIA antibody was raised in rabbits against cortisol-3-CMO; Bovine serum albumin was used as a blocking agent and cortisol-3-CMO linked to diamino-3,6-dioxaoctane-biotin was used as label. Microtiter plates were coated with anti-sheep-IgG (1 lg/well). The antibody working dilution was 1:100.000; and the label was diluted 1:500.000. The standard curve was between 0.33–80 pg cortisol. Intra- and inter-assay coefficients of variation of high and low value quality controls were 8.6 and 14.5% (*n* = 37) and 10.4 and 14.1% (*n* = 93), respectively. All samples were measured in duplicates. Samples were re-measured in an adequate dilution if bindings were outside the linear range of the assay, or if divergence of duplicates was greater than 10%. A total of 3560 salivary immunoreactive cortisol measures were entered into the dataset.

### 2.5. Data Preparation and Statistical Analyses

#### 2.5.1. Estimating Salivary Cortisol Slopes and Intercepts

First, we fitted linear mixed models with Gaussian error structures, e.g., [58], to calculate individuals’ cortisol intercepts and diurnal slopes by condition (i, ii, and iii). We log-transformed the response variable salivary cortisol levels to achieve a more symmetrical distribution. The models were controlled for potential effects of the sex of individual subjects and for age-related differences in salivary cortisol patterns, since sex effects had been noted for bonobos (but not orangutans or gorillas [44]), and since with increasing age, cortisol levels are elevated and the circadian decline is flatter [59,60]. The control variable age was centered to a mean of zero with a standard deviation of one (z-transformation). We added a random intercept for the individual and a random slope for the time of day a sample was collected (z-transformed to obtain better estimates) to the models (one model for each condition). From these models, we extracted intercept and diurnal slope estimates for each individual and condition. Models were fitted in RStudio, Version 1.3.959; RStudio, Inc. in Boston, Massachusetts, USA [61] using the function “lmer” (R package “lme4”, [62]).

#### 2.5.2. Building Models to Test Hypotheses

The extracted random effect values per individual and per condition were then used as response variables in the linear models [63] built to test our hypotheses. Similar to the models above, in all models, the control variable age was averaged over the duration of each condition (e.g., average age in days across the time an individual was housed in the old building) and then z-transformed before being entered into the linear models. Sex was entered as a control factor in all models described below.

We fitted two linear models with (i) individuals’ salivary cortisol intercepts (“intercept model”); and (ii) the individuals’ salivary diurnal cortisol slopes (across sample collection times per collection day) (“slope model”) as response variable. Visual inspection suggested a nearly normal distribution of both variables. To test for potentially species-specific effects of each condition on both models, we entered the interaction between condition and species as a predictor term in each of them.

We established significance of the full model [64] using a likelihood ratio test [65] by comparing it with the respective null model containing only the control variables. If full and null model did not differ significantly from each other (threshold for statistical significance set at *p* = 0.05), the simpler model (null model) was considered the final model. In case the two models were significantly different, we reduced model complexity of the full model by eliminating interaction terms (one at a time), and keeping main effects of individual variables. Likelihood ratio tests were used to compare competing models with each other (i.e., model containing the two-way interaction with the simplified model containing main effects). We checked linear relationship assumptions, residual distributions, homoscedasticity, and tested for influential cases by visual inspection of diagnostic plots to ensure models did not violate underlying model assumptions.

## 3. Results

Models included 78 estimates of individual SCI and SCS (Appendix A). Salivary cortisol means, medians, and ranges are presented for each sex and species in Table 2.

SCI ranged from −0.37 to 0.79 across all species and conditions, and SCS varied between −1.01 and 0.69. In our study, 18% (*n* = 14) of the slopes were positive. Nine of the cases occurred in bonobos, where the adult male bonobo had two positive slopes. Among gorillas, an old female had the highest number of positive slopes (*n* = 2). In orangutans, no individual had positive slopes. Overall, only two of the positive slopes were found in the old house. Five outliers of two gorilla females were excluded from the dataset.

When looking at SCI, we found that the full intercept model and the respective null model differed significantly from each other (Chi-square = −0.80, df = 8, *p* < 0.001). Including the interaction term between species and condition was not significantly different to a model containing only the main effects (Chi-square = −0.20, df = 4, *p* = 0.106). The final model contained the main effects of species and condition (F(6,71) = 3.85, *p* = 0.00, R2 = 0.25, adjusted R2 = 0.18), both of which significantly predicted SCI (Figure 1; species: Chi-square = −0.469, df = 2, *p* < 0.001; condition: Chi-square = −0.151, df = 2, *p* = 0.05). Post-hoc tests revealed that bonobos had higher SCI than orangutans (estimate ± SE = 0.192 ± 0.045, *p* < 0.001), and gorillas had higher SCI than orangutans (estimate ± SE = 0.121± 0.049, *p* = 0.042). Furthermore, SCI were significantly higher on days individuals were exposed to enrichment in comparison to days without enrichment in the old building (estimate ± SE = 0.107 ± 0.045, *p* = 0.05). No other significant differences between conditions were found (enrichment day versus new ape house: estimate ± SE = 0. 0.065 ± 0.046, *p* = 0.346; new versus old ape house: estimate ± SE = 0.042 ± 0.045, *p* = 0.615).

The SCS full model was significantly different from its null model (Chi-square = −3.01, df = 8, *p* < 0.01). The final model contained species and condition as main effects, after eliminating the interaction term to reduce model complexity (F(6,71) = 6.7, *p* = 0.00, R2 = 0.36, adjusted R2 = 0.31). Condition (Chi-square = −1.42, df = 2, *p* < 0.01) and species (Chi-square = −1.14, df = 2, *p* < 0.01) significantly predicted SCS (Figure 2). Post-hoc comparisons revealed that slopes of individuals of all three species were significantly steeper in the old enclosures than after being transferred to the new ape building (estimate ± SE = 0.326 ± 0.07, *p* < 0.01) (Figure 2). No difference was found between salivary cortisol slopes on enrichment days and either housing condition (enrichment-old house: estimate ± SE = 0.117 ± 0.07, *p* = 0.222; enrichment-new house: estimate ± SE = −0.209 ± 0.072, *p* = 0.133). Both, gorillas (estimate ± SE = 0.174 ± 0.069, *p* = 0.036) and orangutans (estimate ± SE = 0.291 ± 0.071, *p* < 0.01) had steeper slopes than bonobos. Slope steepness did not differ between gorillas and orangutans (estimate ± SE = 0.117 ± 0.076, *p* = 0.274).

## 4. Discussion

Our study is the first to use a reaction norm approach, quantifying SCI and SCS in response to changes in the environment in zoo-housed great apes. We show that changes in SCI and SCS are independent from each other: an increase in SCI is not necessarily associated with a blunted or steeper SCS. This is in line with other studies investigating phenotypic flexibility and reaction norms of stress physiology [32,66]. Conditions in the old ape house can be characterized as highly predictable to the apes, and thus apes probably perceived them as having a high degree of control [51]. Therefore, SCI and SCS in this environment can be considered as “baseline” patters of cortisol levels. Days in the old ape house are characterized by salivary cortisol levels varying within a certain range, as individuals respond to everyday challenges and maintain vital functions (e.g., metabolic processes) and show a steady diurnal salivary cortisol decrease throughout the day. One potential factor that might impact our results is that the first gorilla samples were collected earlier (11 a.m.) in the day than samples of the two other species in our study (1 p.m.). However, the salivary cortisol level decline in bonobos and chimpanzees between 11 a.m. and 1 p.m. is relatively small. As the intercept of the curve is the constant with which cortisol levels decrease, and the decrease is assumed to be steady, the bias introduced in our data due to difference in the timing of sample collection should be relatively mild. For both, the estimates of SCI and SCS, a major concern would have been if the steep decline in cortisol levels associated with the awakening response [23,67] would have been included in our sample, which is not the case though. However, future studies should ideally keep sampling conditions constant across species to avoid potential difficulties in the interpretation of results.

The three ape species investigated in our study differed from each other in regards to their SCI (bonobos had higher intercepts than gorillas, who had higher intercepts than orangutans), but no species differences were found in SCS. Although the bias introduced by differences in sampling regimes is mild (see above), it is difficult to conclude that the differences in SCIs between the species are truly species-specific, because we only have one group per species. Thus, the differences could be caused by group effects rather than species effects. However, differences in great ape salivary cortisol measurements were already reported earlier including different groups of different zoos [44]. These differences can be explained by various factors, ranging from disparities in their sociality and aggression rates, e.g., [68], feeding competition, e.g., [69], or merely potential differences in cortisol receptor density (for instance in New World primates [70]). Future studies could focus on larger sample sizes to investigate within or between species differences in SCI.

Crucially, in the “baseline” condition (old house), all three ape species had a steady decrease in salivary cortisol levels throughout the day (negative SCSs), showing the expected decline in circadian cortisol excretion [18,24,25,27,71]. Only three slopes were positive in the old house. On enrichment days, salivary cortisol excretion patterns changed in all three ape species. The SCI increased compared to the old house condition, but SCS did not differ from the old house condition. The fact that SCI increased on enrichment days, is in line with other studies in apes, which showed increased glucocorticoid levels in response to environmental enrichment [53,54]. However, combining estimates of SCI and SCS shows that, although the HPA axis responded to the disruption of the homeostasis (as elicited by the enrichment) with an increase in cortisol levels, the typical decline in cortisol excretion was preserved (as indicated by comparable SCS in the old house and on enrichment days). This suggests that the stressor (enrichment) resulted in a short-term stress response, but the diurnal salivary cortisol excretion rates were not affected. Enrichment in zoo-housed individuals can be regarded as an aspect in their environment they can control, which may be the reason why the stress experience in such contexts is reduced [72]. Rewarding stimuli, such as enrichment items, can be associated with a higher stress response than negative stimuli. For example, appetitive situations like victory or sexual activity are associated with higher glucocorticoid responses than footshock or handling in rats [51]. Despite an increase in cortisol levels or precisely because of this rise that is accompanied by a SCS, typical for a healthy HPA functioning, enrichment improves animal physiological and psychological well-being [73,74,75].

For all three ape species, the SCI in the new ape house were higher than in the old ape house, and lower than on enrichment days, although the differences were not significant. However, the SCS of individuals when housed in the new ape enclosures were blunted, indicating that cortisol levels did not decline following the normal diurnal pattern. Rather, salivary cortisol levels stayed elevated or increased even more, as expected for individuals, who cope with a persistent stressor [76]. Previous studies have shown that a new environment [53,54] or a more enriched environment can elevate glucocorticoid levels for extended periods of time—stretching even over months [41,77,78]. In comparison to the old ape house, the SCS in the new ape house were less steep, but the SCI were not particularly high. This pattern of flat SCS but not particularly high SCI suggests that the apes expressed a persistent stress response of low magnitude. Such a response may allow individuals to cope with a novel situation, and maybe could be categorized as a tolerable stressor [43]. For a better understanding of a new environment as a stressor, future studies need to collect more samples of each individual after the transfer. This would allow us to estimate the time a species or an individual need to adapt or cope with a new situation. The variation in those processes might be related to individual coping styles, which are part of individual personalities [41,79].

One important limitation in our study is the sampling time. For several practical reasons we chose, both in the old house and in the new house, to collect saliva samples only in the afternoon with a three-to-four-hour time window. First of all, we wanted to avoid interference of feeding time and food consumption on cortisol measures. Second, it was more practical to collect the samples in the afternoon, because then sampling interfered less with keeper routines in the morning. Third, the effects of providing enrichment in the second part of our study were believed to be greatest in the afternoon, because in the morning, apes would be engaged in collecting food from their breakfast and exploring the enclosure after morning cleaning routines, etc. This approach means, however, that we did not include samples from the early morning or late afternoon as done in human studies of cortisol slopes [10,80]. Previous studies in chimpanzees [67], bonobos [23], and orangutans [52] have demonstrated that also in these species, there is a peak in salivary cortisol in early morning (depending a bit on when samples were taken 6 a.m. in chimpanzees, 8:30 in bonobos), followed by a steep decline until noon, and then followed by a slower decline between noon and 4 p.m. with nearly half of the level at 4 p.m. compared to noon [23,67]. This means in our study we only measured the slow decline during the afternoon. However, for the following reasons, we are confident that our data are a good proxy for diurnal cortisol patterns. First, most estimated slopes show a decrease in cortisol levels from the first to the second sample within a given day. If short-term fluctuations in cortisol levels would override the diurnal decrease in cortisol, we would expect that the slopes in our study slopes would not show such a uniform pattern in the expected direction. Second, we found this pattern in all three species, despite the difference in time between samplings. If anything, we would have expected to not find any patterns in slope and intercept if the samplings were to close together for a meaningful difference between samples. Nevertheless, we encourage future studies on diurnal slopes of salivary cortisol in captive apes to consider to collect early and late samples and more samples in between.

The welfare of animals under human care is becoming an increasingly important topic for both science and society, as evidenced by the numerous ethical debates currently taking place in public. With the commitment to improve animal welfare, comes the need for objective, non-invasive means to measure and quantify it [81,82,83]. Human stress research but also ecological studies use reaction norms of labile traits (such as cortisol measures) to assess the state of (human or population) well-being [84,85]. In animal welfare assessment, this simple tool to look at physiological parameters is not used so far, to our knowledge. Changes in cortisol alone, or single event sampling, do not allow us to draw overall conclusions on animals’ stress levels or welfare [4,14,77]. For example, the associations of cortisol levels with and without environmental enrichment were diverse: while enrichment increased glucocorticoid levels in some studies (*Pan troglodytes* [53], *Pongo pygmaeus* [54]), levels declined during enrichment programs in other studies (*Pongo pygmaeus* [52], *Xhrysocyon brachyurus* [74], *Tamandua tetradactyla* [86], *Leopardus tigrinus* [87], *Sebastes schlegelii* [88]), or were found to be unaffected (*Panthera tigris tigris* and *Felis concolor* [75], *Leopardus wiedii* [87]). These contrasting results indicate that mere cortisol levels are not an ideal indicator of an animal’s welfare or stress state. Absolute cortisol measures may indicate an activation of the HPA axis, but do not allow for assumptions about the quality of the stressor disrupting homeostasis (short-term vs. persistent), or for the valence of the arousal (positive or negative) [14]. Another aspect to consider is that in our study the stress responses (changes in the SCI) were species-specific, whereas changes in SCS were independent of the species. This could be a pattern that holds across many species and conditions, as the circadian cortisol rhythm is such a widespread and vital phenomenon.

Finally, we like to emphasize that looking at salivary cortisol reaction norms only sheds light on one aspect of the stress response, which itself is not only composed of the HPA axis activity, but also includes activation of the Autonomic Nervous System, and that the stress response is only one of many aspects relevant to animal welfare. In several species, a combination of different stress or welfare markers are being used to monitor the state of individuals. This may be a useful approach also in great apes that is currently implemented at diverse sites such as zoos. A set of several biomarkers that can be measured in saliva samples, may explain stress responses better than each marker alone [89]. A candidate here is for instance to look at the ratio of salivary alpha amylase (a proxy for the activation of the sympathetic nervous system, which has already been measured in great apes [44]) to cortisol. Various other physiological markers potentially relevant in an animal welfare context are for example, dehydroepiandrosterone (-sulfate) (DHEA(-S)) [90,91], or the ratio of DHEA/cortisol as indicator of immune function, mental health, cognitive performance, and overall welfare [91,92]. Another aspect that should be considered is that welfare assessment is more than the measurement of physiological markers. Behavioral observations can pinpoint the level of an individual’s social integration, self-directed behaviors, or stereotypies that are crucial indicators of animal welfare.

## 5. Conclusions

Our study provides evidence that SCI in combination with changes in SCS can differentiate between types of stress responses, and thereby constitute one useful measure for welfare assessment. We conclude that if saliva collection is feasible in a species twice per day, tracking changes in SCI in combination with SCS provides a possibility of stress response quality assessment, which might be a useful tool in animal welfare assessment. While physiological markers offer a good possibility to monitor physiological changes of an individual, results should always be interpreted with caution, within the scope of the broader context, and ideally in combination with other relevant markers and parameters.

## Figures and Tables

**Figure 1 animals-12-00522-f001:**
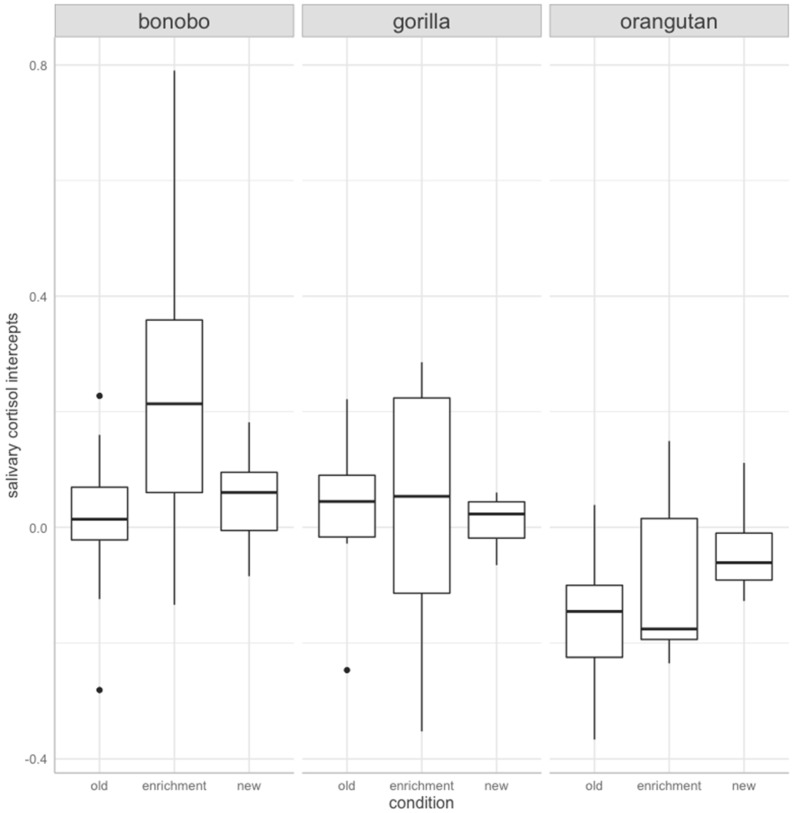
Salivary cortisol intercepts by condition and species. Individuals of all three species (bonobo, gorilla, and orangutan) had significantly higher salivary cortisol intercepts on enrichment days as compared to the other two conditions. Across conditions, bonobos’ salivary cortisol intercepts were significantly higher in comparison to gorillas and orangutans, and gorillas were also significantly higher than orangutans.

**Figure 2 animals-12-00522-f002:**
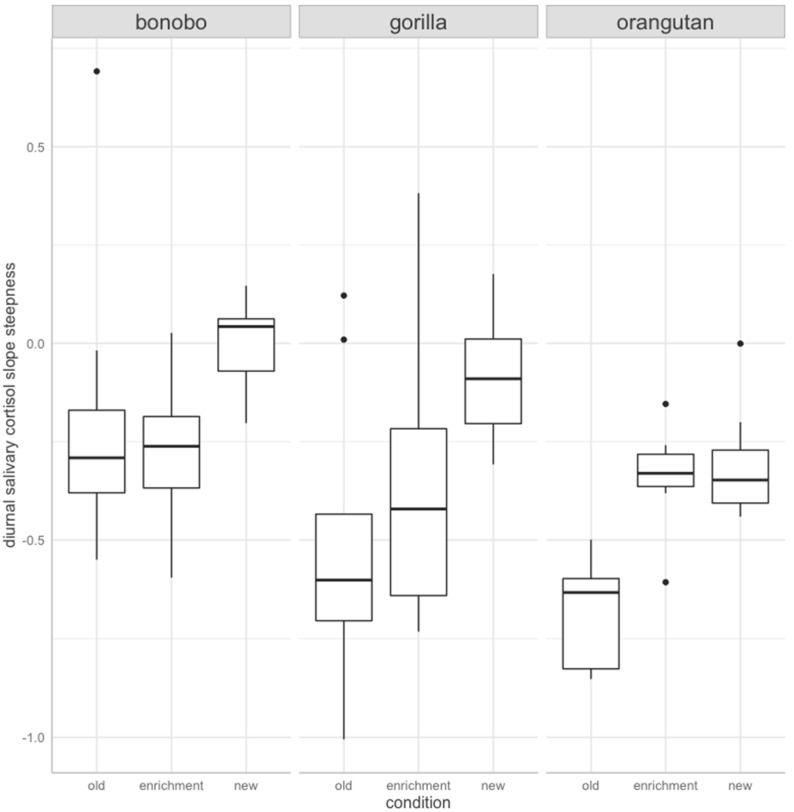
Salivary cortisol slopes by condition and species. On the *y*−axis steepness of the slope is indicated: negative values indicate decreases in salivary cortisol levels across the day: values greater than zero, indicate an increase in salivary cortisol levels across the day. The closer an estimate is to zero, the flatter is the slope. Individuals of all three species (bonobo, gorilla, and orangutan) had significantly steeper salivary cortisol slopes on routine days in the old building than on routine days after the transfer to the new building. Salivary cortisol slopes on enrichment days (in the old building only) did not differ significantly from either slopes on routine days in the old building or routine days in the new building. Enrichment days were associated with intermediate salivary cortisol slope steepness across the day.

**Table 1 animals-12-00522-t001:** Number of days of data collection; total number of samples; average, maximum and minimum number of samples per individual (ID) per species and per study section.

Species	Parameter	Old Ape House	Enrichment Days	New Ape House
Bonobo	Days	51	12	25
	Total samples	852	414	441
	Average sample/ID	71.4	13.8	40.1
	Max sample/ID	94	16	48
	Min sample/ID	9	6	7
Gorilla	Days	45	12	22
	Total samples	453	133	125
	Average sample/ID	50.3	5.8	17.9
	Max sample/ID	74	8	32
	Min sample/ID	6	2	10
Orangutan	Days	38	12	31
	Total samples	499	304	339
	Average sample/ID	71.3	14.5	48.4
	Max sample/ID	73	16	54
	Min sample/ID	66	7	35

**Table 2 animals-12-00522-t002:** Salivary cortisol (ng/mL) means, medians, ranges, and standard deviations per species, sex, condition, and sampling time.

Species	Sex	Sampling Timw	Variable	Old Ape House	Enrichment Days	New Ape House
				Salivary Cortisol (ng/mL)		
Bonobo	Female	First sample	Mean	4.7	6.3	6.3
			Median	3.9	5.5	5.4
			Range	0.4–20.6	2.2–18.6	0.7–28.1
			SD	3.5	3.3	3.9
			*n*	280	76	146
		Last sample	Mean	3.3	4.0	6.2
			Median	2.5	3.5	5.0
			Range	0.3–20.3	1.2–16.6	0.9–32.2
			SD	2.4	2.2	4.4
			*n*	293	73	152
	Male	First sample	Mean	4.9	6.6	7.9
			Median	4.1	4.4	6.6
			Range	0.6–15.2	1–20.7	0.5–36.9
			SD	3.1	5.1	5.6
			*n*	140	37	70
		Last sample	Mean	4.2	5.9	6.3
			Median	3.5	4.3	5.3
			Range	0.5–14.6	0.7–18.5	0.6–26.9
			SD	2.6	4.5	4.4
			*n*	144	32	73
Gorilla	Female	First sample	Mean	7.6	5.9	6.6
			Median	4.5	5.8	6.3
			Range	0.3–203	0.2–20	0.9–18.3
			SD	18.0	3.8	4.0
			*n*	192	48	39
		Last sample	Mean	4.9	4.3	5.9
			Median	1.7	2.5	5.0
			Range	0.1–245	0.5–32.8	1.4–13.1
			SD	20.7	6.3	3.2
			*n*	146	48	31
	Male	First sample	Mean	7.9	6.4	6.0
			Median	5.2	6.4	4.7
			Range	1.3–49.2	1.5–13.3	1.1–25.6
			SD	9.9	3.6	4.8
			*n*	59	18	33
		Last sample	Mean	1.9	2.4	4.2
			Median	1.4	1.8	3.5
			Range	0.2–10.5	0.3–6.7	1.0–8.1
			SD	1.9	1.7	2.1
			*n*	54	19	22
Orangutan	Female	First sample	Mean	2.8	3.6	4.3
			Median	2.3	3.1	3.5
			Range	0.6–25.7	1.3–16.0	0.3–19.5
			SD	2.5	2.4	3.0
			*n*	140	45	98
		Last sample	Mean	1.8	3.0	3.4
			Median	1.2	2.2	2.7
			Range	0.2–34.3	0.9–15.2	0.3–12.7
			SD	3.0	2.5	2.6
			*n*	143	40	100
	Male	First sample	Mean	2.8	3.6	4.4
			Median	2.6	3.5	4.0
			Range	0.7–6.1	0.1–9.3	0.9–12.6
			SD	1.2	1.6	2.2
			*n*	107	36	73
		Last sample	Mean	2.3	2.7	3.4
			Median	1.6	2.3	3.0
			Range	0.1–35.1	0.6–8.9	0.8–12.1
			SD	3.7	1.5	2.0
			*n*	109	35	68

## Data Availability

“Replication Data for reaction norm approach in salivary cortisol of zoo housed apes” Available online: https://doi.org/10.25625/KW3DVD (accessed on 21 December 2021) and is available upon request.

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
