# Peer review of "Salivary Cortisol Reaction Norms in Zoo-Housed Great Apes: Diurnal Slopes and Intercepts as Indicators of Stress Response Quality"

_animals, 2022, doi:10.3390/ani12040522_

Round 1
Reviewer 1 Report
The authors are congratulated on submitting a well written manuscript discussing the difficulties and challenges to interpret cortisol measurements in regards to stress. They manage to scrutinize the common practice of interpreting one event-one cortisol measurement by offering and elaborating on an alternative method, without depicting it as the ultimate truth. Their modesty only underlines their professional and innovative approach to "stress" measurement and analysis .
I haven't had many manuscripts with only that little to criticize. The manuscript is very well written and carefully brings an uninformed reader to easily understand the topic and method.
There are a few text editing suggestions that I added in the manuscript (see attachment), otherwise I definitely consider this a valuable contribution to the journal.

Author Response
We would like to thank Reviewer 1 for the kind words and the positive feedback on our work. We have implemented all suggested edits.
- We now have changed “adapted” to “adapt” (in the manuscript version with track changes line 198)
- We now deleted “after a sufficient amount of saliva was absorbed by the swab” (Lines 209-210)
- We re-worded: We fitted two linear models with (i) individuals’ salivary cortisol intercepts (“intercept model”); and (ii) the individuals’ salivary diurnal cortisol slopes (across sample collection times per collection day) (“slope model”) as response variable. (Lines 272-274)
- Following a suggestion by another reviewer we have now changed “because to” to “due to” (Line 306)
- We now changed “different” to “differed”. (Line 310)
Reviewer 2 Report
This is an interesting paper that tests a new and creative way of measuring reaction norms in cortisol in zoo animals. I think this represents an important advance in our knowledge of how to assess 'stress' and this method could be applied in our contexts - in both captivity and in the wild. Thus it is of general interest.
I have a few comments:
The manuscript needs to be more explicit in saying that cortisol is not a measure of 'stress' in the summary and the abstract. Later in the manuscript they show that they have a more nuanced understanding of the function of cortisol, but the way it is presented in the summary and abstract reinforces an over simplified view of this hormone. For example, the study finds that cortisol levels increase with enrichment. This is, correctly, not interpreted as the animals being more 'stressed.' It's important to educate about what cortisol is telling us - that it is about energy activation - which could be from psychological distress but also in response to food intake or just general excitement and increased energy expenditure.
The fact that the three species showed the same general pattern is very interesting (Figure 2). Figure 1 shows that there appear to be species differences. I would be cautious with this interpretation because of the small sample sizes. The groups also differed by other factors beyond species: how long the individuals were in that group, group dynamics, whether the animals were related, etc. You should encourage people to repeat this study to see if they get the same species differences and to see if it is really 'species' that accounts for the differences or some other factor that differed between the groups.
Specific line corrections. Some of these are suggested changes to the English grammar. There are other sentences where I just note that the sentence structure or wording is awkward and needs to be rewritten to be clear.
Line 42: add "us" to differentiate
Line 48: Should be "For" not "Since"Line 50: Should be "Gold" not "Golden"
Line 57: Change to "changes in cortisol levels"
Line 62: " can be elicited by all factors that change energy balance". It would be more accurate to say something like, "factors that signal the need to mobilize energy."
Line 83: " approaches to assess the functioning of the diurnal cortisol pattern." This wording is awkward. I think just say, 'assess diurnal cortisol pattern." I guess the question is are you talking about actually figuring out the "Function" of the diurnal pattern or just documenting what the pattern is. It seems like you are just referring to documenting the pattern, and thus don't need to talk about the 'function' of the pattern.
Line 108: May allow "us" to distinguish
Line 112: Comma after magnitude
Line 120: You don't need 'for instance'
Line 121-122: " the examination of diurnal salivary cortisol fluctuations has the potential to inform in a more fine-grained manner about stress responses of individuals and will therefore" This should be reworded.
Line 152: I count six not seven orangutans in your study
Line 174: Should be from 'an' unknown direction
Line 175: "adapt" not "adapted"
Line 176: closed 'to' visitors not 'for'
Line 263: " had with five the highest number of positive slopes." awkward wording
Line 266: "due to" not "because
Line 268: put (with boxes) and (with hoses) in parentheses
Line 270: differed not 'different'
Line 287: Figure 1 legend should be on the same page as the figure.
Line 294: "Reducing model complexity by eliminating the interaction term revealed the model containing species and condition as main effects to be the final model" Awkward sentence structure
Line 321: "as “baseline” of diurnal cortisol excretion pattern" awkward wording
Line 354: Provide a more nuanced explanation that "can be associated with a higher stress response"
Line 372: allow "us"
Line 378: "comes” not came
Line 383: allow "us"
Line 392: Again, avoid oversimplified statements about the 'stress response'
Line 402: monitor 'the' state
Line 403-404: "This may be a useful approach also in great apes that is currently implemented at diverse sites." What 'diverse sites' is this implemented at?
Author Response
We thank the reviewer for the helpful, insightful comments.
The manuscript needs to be more explicit in saying that cortisol is not a measure of 'stress' in the summary and the abstract. Later in the manuscript they show that they have a more nuanced understanding of the function of cortisol, but the way it is presented in the summary and abstract reinforces an over simplified view of this hormone. For example, the study finds that cortisol levels increase with enrichment. This is, correctly, not interpreted as the animals being more 'stressed.' It's important to educate about what cortisol is telling us - that it is about energy activation - which could be from psychological distress but also in response to food intake or just general excitement and increased energy expenditure.
In the summary as well as in the abstract, we added that cortisol changes can be linked to metabolic changes (in the manuscript version with track changes: Lines 16-17and lines 33-34)
The fact that the three species showed the same general pattern is very interesting (Figure 2). Figure 1 shows that there appear to be species differences. I would be cautious with this interpretation because of the small sample sizes. The groups also differed by other factors beyond species: how long the individuals were in that group, group dynamics, whether the animals were related, etc. You should encourage people to repeat this study to see if they get the same species differences and to see if it is really 'species' that accounts for the differences or some other factor that differed between the groups.
To clarify we re-worded in the discussion: The three ape species investigated in our study differed from each other in regards to their SCI (bonobos had higher intercepts than gorillas, who had higher intercepts than orangutans), but no species differences were found in SCS. Although the bias introduced by differences in sampling regimes is mild (see above), it is difficult to conclude that the differences in SCIs between the species are truly species-specific, because we only have one group per species. Thus, the differences could be caused by group effects rather than species effects. However, differences in great ape salivary cortisol measurements were already reported earlier including different groups of different zoos […]. (Lines 385-392)
And
We now added: Future studies could focus on larger sample sizes to investigate within or between species differences in SCI (Lines 401-403)
Specific line corrections. Some of these are suggested changes to the English grammar. There are other sentences where I just note that the sentence structure or wording is awkward and needs to be rewritten to be clear.
We thank the reviewer for the English grammar corrections.
Line 42: add "us" to differentiate
We now added “us” (Line 44).
Line 48: Should be "For" not "Since"
We now changed it to “For” (Line 50)
Line 50: Should be "Gold" not "Golden"
We now changed it to “gold” (Line 52).
Line 57: Change to "changes in cortisol levels"
We now changed it accordingly to: … the interpretation of changes in cortisol levels is not (Line 59).
Line 62: " can be elicited by all factors that change energy balance". It would be more accurate to say something like, "factors that signal the need to mobilize energy."
We now re-worded the sentence to: This adaptive process of restoring homeostasis, resulting in changes in cortisol levels, can be elicited by all factors that signal the need to mobilize energy (Lines 63-65).
Line 83: " approaches to assess the functioning of the diurnal cortisol pattern." This wording is awkward. I think just say, 'assess diurnal cortisol pattern." I guess the question is are you talking about actually figuring out the "Function" of the diurnal pattern or just documenting what the pattern is. It seems like you are just referring to documenting the pattern, and thus don't need to talk about the 'function' of the pattern.
We agree with the reviewers point of view and deleted “the functioning of the” (Line 85).
Line 108: May allow "us" to distinguish
We now added “us” (Line 110).
Line 112: Comma after magnitude
We now added a comma (Line 114).
Line 120: You don't need 'for instance'
We now deleted “for instance” (Line 128)
Line 121-122: " the examination of diurnal salivary cortisol fluctuations has the potential to inform in a more fine-grained manner about stress responses of individuals and will therefore" This should be reworded.
We reworded the sentence: We argue that applying a reaction norm approach to diurnal fluctuations in salivary cortisol can provide qualitative information, useful for better understanding and interpreting individual stress responses - as is already done in humans (Lines 122-124).
Line 152: I count six not seven orangutans in your study
We thank the reviewer for pointing this out, we missed to mention one immature female. Now we have included all seven orangutans. Lines 159-160
Line 174: Should be from 'an' unknown direction
We have now included an “an” before “unknown” (Line 197).
Line 175: "adapt" not "adapted"
We have now changed adapted to adapt (Line 198).
Line 176: closed 'to' visitors not 'for'
We have now changed “for” to “to” (Lines 199-200)
Line 263: " had with five the highest number of positive slopes." awkward wording
We have now re-worded the sentence to: Among gorillas, an old female had the highest number of positive slopes (N = 5). (Lines 302-303)
Line 266: "due to" not "because
We have now changed “because” to “due to” (Line 306).
Line 268: put (with boxes) and (with hoses) in parentheses
We think that would change the meaning, because it was the first enrichment days with boxes and fourth enrichment days with hoses. So only these two days.
Line 270: differed not 'different'
We now changed “different” to “differed” (Line 310)
Line 287: Figure 1 legend should be on the same page as the figure.
We now make sure that the figure legend is on the same page as the figure. Page 11
Line 294: "Reducing model complexity by eliminating the interaction term revealed the model containing species and condition as main effects to be the final model" Awkward sentence structure
We have now re-worded the sentence to: The final model contained species and condition as main effects, after eliminating the interaction term to reduce model complexity. (Lines 335-336)
Line 321: "as “baseline” of diurnal cortisol excretion pattern" awkward wording
We have now re-worded the sentence to: Conditions in the old ape house can be characterized as highly predictable to the apes, and thus apes probably perceived them as having a high degree of control [51]. Therefore, SCI and SCS in this environment can be considered as “baseline” patters of cortisol levels. (Lines 362-365)
Line 354: Provide a more nuanced explanation that ""can be associated with a higher stress response
We have now added an example: Rewarding stimuli, such as enrichment items, can be associated with a higher stress response than negative stimuli. For example, appetitive situations like victory or sexual activity are associated with higher glucocorticoid responses than footshock or handling in rats [51]. (Lines 419-423)
Line 372: allow "us"
We have now added an “us” (Line 440).
Line 378: "comes” not came
We have now changed “came” to “comes” (Line 472).
Line 383: allow "us"
We have now added an “us” (Line 477).
Line 392: Again, avoid oversimplified statements about the 'stress response'
We have now formulated: Absolute cortisol measures may indicate an activation of the HPA axis but do not allow for assumptions about the quality of the stressor disrupting homeostasis (short-term vs. persistent), or for the valence of the arousal (positive or negative) [14]. (Lines 485-488)
Line 402: monitor 'the' state
We have now changed “a” to “the” (line 497)
Line 403-404: "This may be a useful approach also in great apes that is currently implemented at diverse sites." What 'diverse sites' is this implemented at?
We have now added an example: such as zoos. (Line 499)
Reviewer 3 Report
This is an interesting paper with the novel idea of applying salivary cortisol slope and intercept measurements to assess three different environmental conditions in three different species of zoo-housed apes. While this is potentially a valuable approach, I have significant reservations about the experimental methodology and presentation of the results. The methodological concerns are of such a nature as to raise questions about the validity of the authors' findings. These and other issues are discussed below.
- In human studies, salivary cortisol slope and intercept measurements are typically obtained using an early morning (e.g., waking or 1 hour post-waking) and a late day (e.g., evening) sample. As a result, these two parameters are estimated based on cortisol values that are usually quite different (unless the slope is very flat) because of the shape of the diurnal curve and the many hours that have elapsed between collection times. Presumably due to animal management constraints, this was not done here. For the orangutans and bonobos, there was only THREE HOURS between saliva collections at 1 and 4 PM; for gorillas, there was a bit more elapsed time since collections were at 11 AM and 4 PM. The authors need to provide strong justification about how they can obtain valid estimates of cortisol slope and intercept under the conditions of (a) no awakening or other early morning cortisol values, (b) no evening cortisol values, and (c) relatively short time intervals between the two sampling times.
- I'm confused about how the data from the "enrichment" condition were analyzed. Were all salivary four cortisol values used (i.e., 1 PM, 4 PM, and the values at 10- and 20-min post the start of enrichment)? If so, I question whether that is the right approach. If the 10- and/or 20-min cortisol values are higher than the "baseline" at 1 PM (i.e., mild stress response), I don't see how those values can legitimately be used to help construct either the slope or intercept of the diurnal cortisol rhythm. Seems more logical to continue using the first and last samples for the slope and intercept determinations. Moreover, it's not clear whether the animals were already familiar with the enrichment procedures prior to the onset of saliva collection. This should be clarified, because if they weren't then you might expect the cortisol response to presentation of the enrichment items to change over time.
- The data presentation in Tables 1 and 2 leaves much to be desired. Both tables should be expanded considerably. For Table 1, the authors should add information on the range of number of saliva samples across animals of each species under each experimental condition, not simply the total number of samples. For example, for the 12 bonobos in the old ape house condition, what were the smallest and largest number of samples collected per animal when all 12 are considered? For Table 2, merely showing the mean, median, and range of samples per male and female of each species is insufficient. I would like to see the following information presented: for each species, sex, AND experimental condition, what were the mean, median, range, and SD of salivary cortisol concentrations for the FIRST and LAST saliva samples (along with the N's for each value).
- It would be informative to give information about the time course of salivary cortisol levels related to the presentation of the enrichment items. This should include the means, medians, range, and SD of the "baseline", 10-min, and 20-min samples (either in table or figure form). Moreover, those data could be used to determine the magnitude of change from baseline, if a statistically significant change did occur. Such information would add to the existing literature on the physiological effects of enrichment procedures in great ape species.
- Another point seemingly lacking from the data analysis concerns the time course of the response to the move to the new ape house. As far as I can tell, the data analysis simply lumped together all of the salivary cortisol values over time from the "new ape house condition" together in determining the slope and intercept values. Wouldn't it make sense to examine those values over time from early after the move to later on to see whether there is evidence for habituation to the new environment?
- The authors state that their models made use of "290 estimates of individual SCI and SCS". Please clarify the number of animals of each species, sex, and experimental condition within that number of 290.
- A few minor points: (a) If available, please add information on how long it took to obtain the saliva samples from the beginning of the procedure. (b) In section 2.4, there is confusing wording regarding "cross-reactivities". Please clarify. (c) In section 4, lines 339-340, the authors refer to a "steady decrease in salivary cortisol levels throughout the day". How can they possibly claim that, since (except for enrichment days) they only have TWO values obtained several hours apart? Please reword. (d) Also in section 4, the authors refer to circadian cortisol "excretion pattern". This is poor wording, since the circadian rhythm of cortisol reflects a dynamic relationship between time-dependent changes in cortisol output and cortisol metabolism and clearance.
Author Response
- In human studies, salivary cortisol slope and intercept measurements are typically obtained using an early morning (e.g., waking or 1 hour post-waking) and a late day (e.g., evening) sample. As a result, these two parameters are estimated based on cortisol values that are usually quite different (unless the slope is very flat) because of the shape of the diurnal curve and the many hours that have elapsed between collection times. Presumably due to animal management constraints, this was not done here. For the orangutans and bonobos, there was only THREE HOURS between saliva collections at 1 and 4 PM; for gorillas, there was a bit more elapsed time since collections were at 11 AM and 4 PM. The authors need to provide strong justification about how they can obtain valid estimates of cortisol slope and intercept under the conditions of (a) no awakening or other early morning cortisol values, (b) no evening cortisol values, and (c) relatively short time intervals between the two sampling times.
We agree with the reviewer’s opinion that in a perfect setting, samples would have been collected with more time in between samples and at times when salivary cortisol levels can be expected to differ considerably (e.g., early in the morning and late in the afternoon). As the reviewer correctly assumed, this was not possible because of caretaker working hours, zoo opening hours, and animal management regimens such as feeding times, which are known to affect salivary production and cortisol secretion. However, our approach is conservative. We would have been more likely to find differences between the three conditions (old house, enrichment, and new house) if we had had samples from early morning and late afternoon. Thus, the fact that we still find these effects based on samples derived at timepoints when salivary cortisol levels are not expected to differ drastically, strengthens the results. With samples from the early morning and the late afternoon on the other hand, one would have been able to question if the effects still show when the circadian decline in salivary cortisol levels in less pronounced. Furthermore, as we wanted to see the effect of enrichment on cortisol levels, we deliberately chose to provide the enrichment during a time when cortisol levels had already declined to lower levels, and changes due to the enrichment would therefore not be masked by the steep decline of cortisol in the morning [1,2]. For comparative and logistic reasons, the same sampling times then have to be used for the sampling in the other two conditions to allow for a comparison between them.
- Verspeek, J.; Behringer, V.; Laméris, D.W.; Murtagh, R.; Salas, M.; Staes, N.; Deschner, T.; Stevens, J.M.G. Time-Lag of Urinary and Salivary Cortisol Response after a Psychological Stressor in Bonobos (Pan Paniscus). Sci. Rep. 2021, 11, 7905, doi:10.1038/s41598-021-87163-5.
- Heintz, M.R.; Santymire, R.M.; Parr, L.A.; Lonsdorf, E.V. Validation of a Cortisol Enzyme Immunoassay and Characterization of Salivary Cortisol Circadian Rhythm in Chimpanzees (Pan Troglodytes). Am. J. Primatol. 2011, 73, 903–908, doi:10.1002/ajp.20960.
We agree with the reviewer that the sampling times are not optimal for representing intercepts and slopes of the circadian: slopes across the day would have been steeper than estimated based on our samples and intercepts probably would have been higher. Depicting the full circadian rhythm was however not the goal of this study. For the following reasons we are confident that our approach is conservative for testing our hypotheses. First, most estimated slopes show a decrease in cortisol levels from the first to the second sample within a given day. If short-term fluctuations in cortisol levels would override the diurnal decrease in cortisol, we would expect that the slopes in our study would not show such a uniform pattern in the expected direction. Secondly, we found this pattern in all three species, despite the difference in time between samplings. If anything, we would have expected to not find any patterns in slope and intercept if the samplings were too close together for a meaningful difference between samples. Therefore, we are confident that our results still show diurnal changes in cortisol, only missing the extreme decline associated with the early morning elevations in salivary cortisol. We have added a paragraph in the discussion to address these issues. We also described in detail the circadian rhythm in salivary cortisol found in other studies to show that also in the time windrow in our study a decline usually would occur. Of course, we encourage further studies following a different sampling regime that will allow to depict the full diurnal cortisol pattern however.
We have now outlined the limitation of our study One important limitation in our study is the sampling time. For several practical reasons we chose, both in the old house and in the new house, to collect saliva samples only in the afternoon with a three-to-four-hour time window. First of all, we wanted to avoid interference of feeding time and food consumption on cortisol measures. Second, it was more practical to collect the samples in the afternoon because then sampling in-terfered less with keeper routines in the morning. Third the effects of providing en-richment in the second part of our study were believed to be greatest in the afternoon, because in the morning, apes would be engaged in collecting food from their breakfast and exploring the enclosure after morning cleaning routines etc. This approach means however that we did not include samples from the early morning or late afternoon as done in human studies of cortisol slopes [10,80]. Previous studies in chimpanzees [70], bonobos [23] and orangutans [52] have demonstrated that also in these species, there is a peak in salivary cortisol in early morning (depending a bit on when samples were taken 6 am in chimpanzees, 8:30 in bonobos), followed by a steep decline until noon, and then followed by a slower decline between noon and 4pm with nearly half of the level at 4 pm compared to noon [23,70]. This means in our study we only measured the slow decline during the afternoon. However, for the following reasons we are confident that our data are a good proxy for diurnal cortisol patterns. First, most estimated slopes show a de-crease in cortisol levels from the first to the second sample within a given day. If short-term fluctuations in cortisol levels would override the diurnal decrease in cortisol, we would expect that the slopes in our study slopes would not show such a uniform pattern in the expected direction. Second, we found this pattern in all three species, despite the difference in time between samplings. If anything, we would have expected to not find any patterns in slope and intercept if the samplings were to close together for a meaningful difference between samples. Nevertheless, we encourage future studies on diurnal slopes of salivary cortisol in captive apes to consider to collect early and late samples and more samples in between. (In the manuscript with track changes lines 443- 468)
- I'm confused about how the data from the "enrichment" condition were analyzed. Were all salivary four cortisol values used (i.e., 1 PM, 4 PM, and the values at 10- and 20-min post the start of enrichment)? If so, I question whether that is the right approach. If the 10- and/or 20-min cortisol values are higher than the "baseline" at 1 PM (i.e., mild stress response), I don't see how those values can legitimately be used to help construct either the slope or intercept of the diurnal cortisol rhythm. Seems more logical to continue using the first and last samples for the slope and intercept determinations.
We thank the reviewer for this critical comment. Yes, intercepts and slopes of enrichment days were estimated based on all four samples, including the 10 and 20 minute after enrichment exposure samples. We included these samples in order to be comparable to other studies looking at the effect of enrichment on salivary cortisol changes (these often take a sample before enrichment exposure and one sample after or compare samples obtained on enrichment days to samples collected on non-enrichment days). The hypothesis in our work was that despite a surge in salivary cortisol concentrations (which is responsible for the higher intercepts on enrichment days), cortisol levels would then be downregulated, just as is the case on routine days. As this was the main concern of the study, we thought it crucial to actually include the 10 and 20 minutes after enrichment exposure samples. The goal of our work was not so much to monitor the circadian rhythm per se, but rather to use the pattern of the circadian rhythm to interpret changes in intercepts and slopes in relation to our three conditions. To make this clearer we added the following text in our manuscript:
By making use of our knowledge about the pattern of circadian salivary cortisol excretion, we can monitor deviations from this pattern (across the whole day or in specific time windows of the day) in relation to different conditions. (Lines 125-127)
Additionally we already wrote: We also expected to find a typical SCS, similar to the old house pattern, as cortisol levels should be downregulated after the excitement of the enrichment exposure.” (Lines 146-148)
- Moreover, it's not clear whether the animals were already familiar with the enrichment procedures prior to the onset of saliva collection. This should be clarified, because if they weren't then you might expect the cortisol response to presentation of the enrichment items to change over time.
We have now added this information in the method section: All ape species were familiar with these forms of environmental enrichment, as it was provided regularly to the apes prior to this study. During our study, each of these familiar enrichment items was presented to each group on four consecutive days. (Lines 187-190)
- The data presentation in Tables 1 and 2 leaves much to be desired. Both tables should be expanded considerably. For Table 1, the authors should add information on the range of number of saliva samples across animals of each species under each experimental condition, not simply the total number of samples. For example, for the 12 bonobos in the old ape house condition, what were the smallest and largest number of samples collected per animal when all 12 are considered?
We have now added a new table 1 including all the already presented values and in addition the average sample per individual per condition, as well as the min and max sample per individual per condition. Page 5 and 6
For Table 2, merely showing the mean, median, and range of samples per male and female of each species is insufficient. I would like to see the following information presented: for each species, sex, AND experimental condition, what were the mean, median, range, and SD of salivary cortisol concentrations for the FIRST and LAST saliva samples (along with the N's for each value).
We have now prepared a new table 2 including salivary cortisol means, median, ranges and standard deviations with the number of samples shown per species, per sex, per condition, and per sampling time. Page 8 and 9
- It would be informative to give information about the time course of salivary cortisol levels related to the presentation of the enrichment items. This should include the means, medians, range, and SD of the "baseline", 10-min, and 20-min samples (either in table or figure form). Moreover, those data could be used to determine the magnitude of change from baseline, if a statistically significant change did occur. Such information would add to the existing literature on the physiological effects of enrichment procedures in great ape species.
We are grateful for this comment and we like the idea. This is for sure an interesting topic but not within the scope of the present manuscript. We have already run these analyses and plan on preparing a separate manuscript focusing on the impact of different enrichment types on salivary cortisol in these apes.
- Another point seemingly lacking from the data analysis concerns the time course of the response to the move to the new ape house. As far as I can tell, the data analysis simply lumped together all of the salivary cortisol values over time from the "new ape house condition" together in determining the slope and intercept values. Wouldn't it make sense to examine those values over time from early after the move to later on to see whether there is evidence for habituation to the new environment?
We thank the reviewer for outlining this idea, but this would be a paper in itself, running generalized additive mixed models. In the thesis of Behringer ([3]) a rough analysis showed that in gorillas the movement was rising the salivary cortisol levels for a few days, but cortisol levels declined thereafter and stayed stable. In bonobos, salivary cortisol levels stayed elevated for weeks and in orangutans a day, thereafter, in both species’ levels declined and stayed on a stable level.
- Behringer, V. Ethophysiolgische Untersuchung zu haltungsbedingten Einflüssen auf das Verhalten und die Stresssituation von Westlichen Flachlandgorillas (Gorilla g. gorilla), Sumatra Orang-Utans (Pongo abelii) und Bonobos (Pan paniscus) unter Zoobedingungen, Justus-Liebig-Universität: Gießen, 2011.
- The authors state that their models made use of "290 estimates of individual SCI and SCS". Please clarify the number of animals of each species, sex, and experimental condition within that number of 290.
We have now added a Table S1 to the supplement showing in detail the contribution of slopes and intercepts per species per age category and sex.
- A few minor points: (a) If available, please add information on how long it took to obtain the saliva samples from the beginning of the procedure.
We have now added: One person took around seven minutes to collect saliva samples of all orangutans; in the other two species this took marginally longer (Lines 225-226).
- (b) In section 2.4, there is confusing wording regarding "cross-reactivities". Please clarify.
We have now re-structed the sentence for clarification: For immunoreactive cortisol measurement, a cortisol enzyme immunoassay (EIA) was used. The EIA and the cross-reactivities have previously been described by Palme & Möstl [56]. Samples were first diluted 1:10 with assay buffer. (Lines 236-239)
- (c) In section 4, lines 339-340, the authors refer to a "steady decrease in salivary cortisol levels throughout the day". How can they possibly claim that, since (except for enrichment days) they only have TWO values obtained several hours apart? Please reword.
We have now re-worded to: Crucially, in the “baseline” condition (old house), all three ape species had a steady decrease in salivary cortisol levels throughout the day (negative SCSs), showing the expected decline in circadian cortisol excretion [18,24,25,27,71]. (Lines 404-406)
- (d) Also in section 4, the authors refer to circadian cortisol "excretion pattern". This is poor wording, since the circadian rhythm of cortisol reflects a dynamic relationship between time-dependent changes in cortisol output and cortisol metabolism and clearance.
We have already changed the sentence as suggested by another reviewer to: Conditions in the old ape house can be characterized as highly predictable to the apes, and thus apes probably perceived them as having a high degree of control [51]. Therefore, SCI and SCS in this environment can be considered as “baseline” patters of cortisol levels. (Lines 362-365)
Reviewer 4 Report
Summary
Overall, good, and presented in a well-structured manner. The topic of measuring reaction norm salivary cortisol intercepts and reaction norm salivary cortisol slopes is an interesting subject that has implications for understanding zoo animal behaviour and management and it is in line with the Journal aims and scope.
Introduction
The introduction is well-defined, the cited references are adequate and don’t include an abnormal number of self-citations. The sections are well-developed, and the hypothesis clearly presented.
Materials and Methods
I would suggest providing more information on exhibit types to better understand how they impact on cortisol levels, as no behavioural data were collected.
In the “Saliva sampling protocol” (line numbers 190-192) authors state that: due to management reasons, saliva sample collection from gorillas was undertaken at 11 am and at 4 pm in all three conditions. Could they explain how this difference in the sampling protocol among the study species affected the results?
The sampling period is not clear. Were all the samples from all the species in the Old ape house and during the enrichment days collected in November 2006 (line number 185)? How is it possible if the samples were collected, for example for Bonobos, for 52+12 days (Table 1)? Were the samples in the New ape house collected in December 2008 (line number 185)? If all apes were moved to the new enclosure in May 2008 (line number 171), why wait so long before making the sampling? Could authors explain it better?
The amount of saliva samples collected was adequate, and I am confident that there was enough sampling done to address the objectives of the authors.
The overall experimental design is appropriate to test the hypothesis.
Statistical analysis has been performed appropriately.
Results
Results are clearly presented, and figures and tables are easy to interpret.
In “Sample preparation and analytical methods” (line number 209) the range of the standard curve was expressed in pg, while in the results (Table 2) in ng/mL. I would suggest using the same unit of measure.
Discussion and conclusions
The manuscript scientifically sound, data are appropriately interpreted throughout the manuscript and support the conclusions.
Ethics statements are adequate as all saliva samples were obtained using non-invasive methods.
Data availability statements are adequate as authors made all data underlying the findings in their manuscript available.
References
I would suggest checking the references formatting as sometimes the journal names are written in full and sometimes are abbreviate. For example:
3. Ali, N.; Nater, U.M. Salivary Alpha-Amylase as a Biomarker of Stress in Behavioral Medicine. Int.J. Behav. Med. 2020, 27, 337– 342, doi:10.1007/s12529-019-09843-x.
22. Menargues, A.; Urios, V.; Limiñana, R.; Mauri, M. Circadian Rhythm of Salivary Cortisol in Asian Elephants (Elephas maximus): A Factor to Consider during Welfare Assessment. Journal of Applied Animal Welfare Science 2012, 15, 383–390, doi:10.1080/10888705.2012.709157.
Author Response
I would suggest providing more information on exhibit types to better understand how they impact on cortisol levels, as no behavioural data were collected.
We have now added additional information about enclosure sizes and structure in material and method part (In the manuscript with track changes lines 171-182 and 201-205).
In the “Saliva sampling protocol” (line numbers 190-192) authors state that: due to management reasons, saliva sample collection from gorillas was undertaken at 11 am and at 4 pm in all three conditions. Could they explain how this difference in the sampling protocol among the study species affected the results?
The effect of differences in the sampling protocol are now discussed: One potential factor that might impact our results is that the first gorilla samples were collected earlier (11 am) in the day than samples of the two other species in our study (1 pm). However, the salivary cortisol level decline in bonobos and chimpanzees between 11 am and 1 pm is relatively small. As the intercept of the curve is the constant with which cortisol levels decrease, and the decrease is assumed to be steady, the bias introduced in our data due to difference in the timing of sample collection should be relatively mild. For both, the estimates of SCI and SCS, a major concern would have been if the steep decline in cortisol levels associated with the awakening response [23,70] would have been included in our sample, which is not the case though. However, future studies should ideally keep sampling conditions constant across species to avoid potential difficulties in the interpretation of results. (Line 374-384)
The sampling period is not clear. Were all the samples from all the species in the Old ape house and during the enrichment days collected in November 2006 (line number 185)? How is it possible if the samples were collected, for example for Bonobos, for 52+12 days (Table 1)? Were the samples in the New ape house collected in December 2008 (line number 185)? If all apes were moved to the new enclosure in May 2008 (line number 171), why wait so long before making the sampling? Could authors explain it better?
We have now added: Samples were collected between November 2006 and December 2008. In the old ape house, samples were collected between November 2006 and May 2008. Within this period enrichment was provided and sampling of saliva on enrichment days took place. The apes were moved in May 2008 to their new ape houses. Saliva was then collected in the new ape house from May 2008 until the end of December 2008. (Lines 213-217)
In “Sample preparation and analytical methods” (line number 209) the range of the standard curve was expressed in pg, while in the results (Table 2) in ng/mL. I would suggest using the same unit of measure.
We thank the reviewer for pointing this out, however, the standard curve is in pg and for the samples we used a dilution to be in the range of the curve. In the final calculation we multiply by the dilution factor and end up with ng/mL.
I would suggest checking the references formatting as sometimes the journal names are written in full and sometimes are abbreviate
We have now carefully checked the references.
Round 2
Reviewer 3 Report
I thank the authors for their careful attention to the concerns raised in my review. As a result of the changes I requested (re: justification and clarification of the methods, more detailed presentation of the results, and discussion of the study limitations in the manuscript) and those requested by the other reviewers, the manuscript is significantly improved. However, now that the data are more fully presented, it was a major surprise to find that, as shown in the supplementary table, only a SINGLE slope and intercept per animal were used analytically for the old and new ape house conditions, whereas up to TEN values were calculated per animal for the enrichment condition. This discrepancy is despite the fact that many more saliva samples were obtained in the old and new ape house conditions compared to enrichment days. If the single values for old and new ape house conditions represent MEAN slopes and intercepts, that would be reasonable but it should be clarified somewhere (perhaps it is in the statistics section, though that section is rather dense to read). Yet, it still remains to be clarified why there are multiple (up to 10) slopes and intercepts per animal used for the enrichment condition. Please explain all of this in the manuscript.
Author Response
We are really grateful for the reviewer’s comment. We now have re-run the analyses. All results stay the same. We modified the results section as well as the figures accordingly.
Line: 280
Lines: 287-306
Figure 1
Lines 317-328
Figure 2
As well as suppl. Table 1